# Mixed Precision Quantization of ConvNets via Differentiable Neural Architecture Search

## Abstract

Recent work in network quantization has substantially reduced the time and space complexity of neural network inference, enabling their deployment on embedded and mobile devices with limited computational and memory resources. However, existing quantization methods often represent all weights and activations with the same precision (bit-width). In this paper, we explore a new dimension of the design space: quantizing different layers with different bit-widths. We formulate this problem as a neural architecture search problem and propose a novel differentiable neural architecture search (DNAS) framework to efficiently explore its exponential search space with gradient-based optimization. Experiments show we surpass the state-of-the-art compression of ResNet on CIFAR-10 and ImageNet. Our quantized models with 21.1x smaller model size or 103.9x lower computational cost can still outperform baseline quantized or even full precision models.

## 1 Introduction

Recently, ConvNets have become the *de-facto* method in a wide range of computer vision tasks, achieving state-of-the-art performance. However, due to high computation complexity, it is non-trivial to deploy ConvNets to embedded and mobile devices with limited computational and storage budgets. In recent years, research efforts in both software and hardware have focused on low-precision inference of ConvNets. Most of the existing quantization methods use the same precision for all (or most of) the layers of a ConvNet. However, such uniform bit-width assignment can be suboptimal since quantizing different layers can have different impact on the accuracy and efficiency of the overall network. Although mixed precision computation is widely supported in a wide range of hardware platforms such as CPUs, FPGAs, and dedicated accelerators, prior efforts have not thoroughly explored the mixed precision quantization of ConvNets.

For a ConvNet with $N$ layers and $M$ candidate precisions in each layer, we want to find an optimal assignment of precisions to minimize the cost in terms of model size, memory footprint or computation, while keeping the accuracy. An exhaustive combinatorial search has exponential time complexity ($\mathcal{O}(M^N)$). Therefore, we need a more efficient approach to explore the design space.

In this work, we propose a novel, effective, and efficient differentiable neural architecture search (DNAS) framework to solve this problem. The idea is illustrated in Fig. 1. The problem of neural architecture search (NAS) aims to find the optimal neural net architecture in a given search space. In the DNAS framework, we represent the architecture search space with a stochastic super net where nodes represent intermediate data tensors of the super net (e.g., feature maps of a ConvNet) and edges represent operators (e.g., convolution layers in a ConvNet). Any candidate architecture can be seen as a child network (sub-graph) of the super net. When executing the super net, edges are executed stochastically and the probability of execution is parameterized by some architecture parameters $\boldsymbol{\theta}$. Under this formulation, we can relax the NAS problem and focus on finding the optimal $\boldsymbol{\theta}$ that gives the optimal expected performance of the stochastic super net. The child network can then be sampled from the optimal architecture distribution.

We solve for the optimal architecture parameter $\boldsymbol{\theta}$ by training the stochastic super net with SGD with respect to both the network's weights and the architecture parameter $\boldsymbol{\theta}$. To compute the gradient of $\boldsymbol{\theta}$, we need to back propagate gradients through discrete random variables that control the stochastic edge execution. To address this, we use the Gumbel SoftMax function (Jang et al. (2016)) to "soft-control" the edges. This allows us to directly compute the gradient estimation of $\boldsymbol{\theta}$ with

a controllable trade-off between bias and variance. Using this technique, the stochastic super net becomes fully differentiable and can be effectively and efficiently solved by SGD.

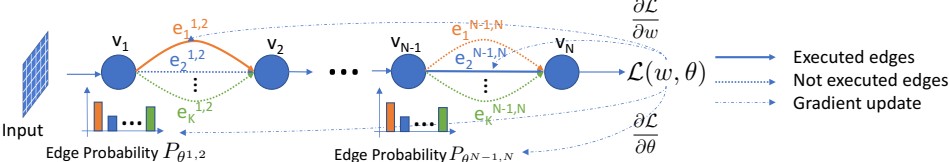

Figure 1: Illustration of a stochastic super net. Nodes represent data tensors and edges represent operators. Edges are executed stochastically following the distribution $P_\theta$. $\theta$ denotes the architecture parameter and $w$ denotes network weights. The stochastic super net is fully differentiable.

We apply the DNAS framework to solve the mixed precision quantization problem, by constructing a super net whose macro architecture (number of layers, filter size of each layer, etc.) is the same as the target network. Each layer of the super net contains several parallel edges representing convolution operators with quantized weights and activations with different precisions. We show that using DNAS to search for layer-wise precision assignments for ResNet models on CIFAR10 and ImageNet, we surpass the state-of-the-art compression. Our quantized models with 21.1x smaller model size or 103.9x smaller computational cost can still outperform baseline quantized or even full precision models. The DNAS pipeline is very fast, taking less than 5 hours on 8 V100 GPUs to complete a search on ResNet18 for ImageNet, while previous NAS algorithms (such as Zoph & Le (2016)) typically take a few hundred GPUs for several days. Last, but not least, DNAS is a general architecture search framework that can be applied to other problems such as efficient ConvNet-structure discovery. Due to the page limit, we will leave the discussion to future publications.

## 2  RELATED WORK

**Network quantization** received a lot of research attention in recent years. Early works such as Han et al. (2015); Zhu et al. (2016); Leng et al. (2017) mainly focus on quantizing neural network weights while still using 32-bit activations. Quantizing weights can reduce the model size of the network and therefore reduce storage space and over-the-air communication cost. More recent works such as Rastegari et al. (2016); Zhou et al. (2016); Choi et al. (2018); Jung et al. (2018); Zhuang et al. (2018) quantize both weights and activations to reduce the computational cost on CPUs and dedicated hardware accelerators. Most of the works use the same precision for all or most of the layers of a network. The problem of mixed precision quantization is rarely explored.

**Neural Architecture Search** becomes an active research area in recent two years. Zoph & Le (2016) first propose to use reinforcement learning to generate neural network architectures with high accuracy and efficiency. However, the proposed method requires huge amounts of computing resources. Pham et al. (2018) propose an efficient neural architecture search (ENAS) framework that drastically reduces the computational cost. ENAS constructs a super network whose weights are shared with its child networks. They use reinforcement learning to train an RNN controller to sample better child networks from the super net. More recently, Liu et al. (2018) propose DARTS, a differentiable architecture search framework. DARTS also constructs a super net whose edges (candidate operators) are parameterized with coefficients computed by a SoftMax function. The super net is trained and edges with the highest coefficients are kept to form the child network. Our proposed DNAS framework is different from DARTS since we use a stochastic super net – in DARTS, the execution of edges are deterministic and the entire super net is trained together. In DNAS, when training the super net, child networks are sampled, decoupled from the super net and trained independently. The idea of super net and stochastic super net is also used in Saxena & Verbeek (2016); Veniat & Denoyer (2017) to explore macro architectures of neural nets. Another related work is He et al. (2018), which uses AutoML for model compression through network pruning. To the best of our knowledge, we are the first to apply neural architecture search to model quantization.

## 3  MIXED PRECISION QUANTIZATION

Normally 32-bit (full-precision) floating point numbers are used to represent weights and activations of neural nets. Quantization projects full-precision weights and activations to fixed-point numbers

with lower bit-width, such as 8, 4, and 1 bit. We follow DoReFa-Net (Zhou et al. (2016)) to quantize weights and PACT (Choi et al. (2018)) to quantize activations. See Appendix A for more details.

For mixed precision quantization, we assume that we have the flexibility to choose different precisions for different layers of a network. Mixed precision computation is widely supported by hardware platforms such as CPUs, FPGAs, and dedicated accelerators. Then the problem is how should we decide the precision for each layer such that we can maintain the accuracy of the network while minimizing the cost in terms of model size or computation. Previous methods use the same precision for all or most of the layers. We expand the design space by choosing different precision assignment from $M$ candidate precisions at $N$ different layers. While exhaustive search yields $\mathcal{O}(M^N)$ time complexity, our automated approach is efficient in finding the optimal precision assignment.

## 4 DIFFERENTIABLE NEURAL ARCHITECTURE SEARCH

### 4.1 NEURAL ARCHITECTURE SEARCH

Formally, the neural architecture search (NAS) problem can be formulated as

$$\min_{a \in \mathcal{A}} \min_{\boldsymbol{w}_a} \mathcal{L}(a, \boldsymbol{w}_a) \tag{1}$$

Here, $a$ denotes a neural architecture, $\mathcal{A}$ denotes the architecture space. $\boldsymbol{w}_a$ denotes the weights of architecture $a$. $\mathcal{L}(\cdot, \cdot)$ represents the loss function on a target dataset given the architecture $a$ and its parameter $\boldsymbol{w}_a$. The loss function is differentiable with respect to $\boldsymbol{w}_a$, but not to $a$. As a consequence, the computational cost of solving the problem in (1) is very high. To solve the inner optimization problem requires to train a neural network $a$ to convergence, which can take days. The outer problem has a discrete search space with exponential complexity. To solve the problem efficiently, we need to avoid enumerating the search space and evaluating each candidate architecture one-by-one.

### 4.2 DIFFERENTIABLE NEURAL ARCHITECTURE SEARCH

We discuss the idea of differentiable neural architecture search (DNAS). The idea is illustrated in Fig. 1. We start by constructing a super net to represent the architecture space $\mathcal{A}$. The super net is essentially a computational DAG (directed acyclic graph) that is denoted as $G = (V, E)$. Each node $v_i \in V$ of the super net represents a data tensor. Between two nodes $v_i$ and $v_j$, there can be $K^{ij}$ edges connecting them, indexed as $e_k^{ij}$. Each edge represents an operator parameterized by its trainable weight $w_k^{ij}$. The operator takes the data tensor at $v_i$ as its input and computes its output as $e_k^{ij}(v_i; w_k^{ij})$. To compute the data tensor at $v_j$, we sum the output of all incoming edges as

$$v_j = \sum_{i,k} e_k^{ij}(v_i; w_k^{ij}). \tag{2}$$

With this representation, any neural net architecture $a \in \mathcal{A}$ can be represented by a subgraph $G_a(V_a, E_a)$ with $V_a \subseteq V, E_a \subseteq E$. For simplicity, in a candidate architecture, we keep all the nodes of the graph, so $V_a = V$. And for a pair of nodes $v_i, v_j$ that are connected by $K^{ij}$ candidate edges, we only select one edge. Formally, in a candidate architecture $a$, we re-write equation (2) as

$$v_j = \sum_{i,k} m_k^{ij} e_k^{ij}(v_i; w_k^{ij}), \tag{3}$$

where $m_k^{ij} \in \{0, 1\}$ is an "edge-mask" and $\sum_k m_k^{ij} = 1$. Note that though the value of $m_k^{ij}$ is discrete, we can still compute the gradient to $m_k^{ij}$. Let $\boldsymbol{m}$ be a vector that consists of $m_k^{ij}$ for all $e_k^{ij} \in E$. For any architecture $a \in \mathcal{A}$, we can encode it using an "edge-mask" vector $\boldsymbol{m}_a$. So we re-write the loss function in equation (1) to an equivalent form as $\mathcal{L}(\boldsymbol{m}_a, \boldsymbol{w}_a)$.

We next convert the super net to a stochastic super net whose edges are executed stochastically. For each edge $e_k^{ij}$, we let $\mathrm{m}_k^{ij} \in \{0, 1\}$ be a random variable and we execute edge $e_k^{ij}$ when $\mathrm{m}_k^{ij}$ is sampled to be 1. We assign each edge a parameter $\theta_k^{ij}$ such that the probability of executing $e_k^{ij}$ is

$$P_{\boldsymbol{\theta}^{ij}}(\mathrm{m}_k^{ij} = 1) = \mathrm{softmax}(\theta_k^{ij} | \boldsymbol{\theta}^{ij}) = \frac{\exp(\theta_k^{ij})}{\sum_{k=1}^{K^{ij}} \exp(\theta_k^{ij})}. \tag{4}$$

The stochastic super net is now parameterized by $\boldsymbol{\theta}$, a vector whose elements are $\theta_k^{ij}$ for all $e_k^{ij} \in E$. From the distribution $P_{\boldsymbol{\theta}}$, we can sample a mask vector $\boldsymbol{m}_a$ that corresponds to a candidate architecture $a \in \mathcal{A}$. We can further compute the expected loss of the stochastic super net as $\mathbb{E}_{a \sim P_{\boldsymbol{\theta}}}[\mathcal{L}(\boldsymbol{m}_a, \boldsymbol{w}_a)]$. The expectation of the loss function is differentiable with respect to $\boldsymbol{w}_a$, but not directly to $\boldsymbol{\theta}$, since we cannot directly back-propagate the gradient to $\boldsymbol{\theta}$ through the discrete random variable $\boldsymbol{m}_a$. To estimate the gradient, we can use Straight-Through estimation (Bengio et al. (2013)) or REINFORCE (Williams (1992)). Our final choice is to use the Gumbel Softmax technique (Jang et al. (2016)), which will be explained in the next section. Now that the expectation of the loss function becomes fully differentiable, we re-write the problem in equation (1) as

$$\min_{\boldsymbol{\theta}} \min_{\boldsymbol{w}_a} \mathbb{E}_{a \sim P_{\boldsymbol{\theta}}}[\mathcal{L}(\boldsymbol{m}_a, \boldsymbol{w}_a)] \tag{5}$$

The combinatorial optimization problem of solving for the optimal architecture $a \in \mathcal{A}$ is relaxed to solving for the optimal architecture-distribution parameter $\boldsymbol{\theta}$ that minimizes the expected loss. Once we obtain the optimal $\boldsymbol{\theta}$, we acquire the optimal architecture by sampling from $P_{\boldsymbol{\theta}}$.

## 4.3 DNAS with Gumbel Softmax

We use stochastic gradient descent (SGD) to solve Equation (5). The optimization process is also denoted as training the stochastic super net. We compute the Monte Carlo estimation of the gradient

$$\nabla_{\boldsymbol{\theta}, \boldsymbol{w}_a} \mathbb{E}_{a \sim P_{\boldsymbol{\theta}}}[\mathcal{L}(\boldsymbol{m}_a, \boldsymbol{w}_a)] \approx \frac{1}{B} \sum_{i=1}^{B} \nabla_{\boldsymbol{\theta}, \boldsymbol{w}_a} \mathcal{L}(\boldsymbol{m}_{a_i}, \boldsymbol{w}_{a_i}), \tag{6}$$

where $a_i$ is an architecture sampled from distribution $P_{\boldsymbol{\theta}}$ and $B$ is the batch size. Equation (6) provides an unbiased estimation of the gradient, but it has high variance, since the size of the architecture space is orders of magnitude larger than any feasible batch size $B$. Such high variance for gradient estimation makes it difficult for SGD to converge.

To address this issue, we use Gumbel Softmax proposed by Jang et al. (2016); Maddison et al. (2016) to control the edge selection. For a node pair $(v_i, v_j)$, instead of applying a "hard" sampling and execute only one edge, we use Gumbel Softmax to apply a "soft" sampling. We compute $\mathrm{m}_k^{ij}$ as

$$\mathrm{m}_k^{ij} = \mathrm{GumbelSoftmax}(\theta_k^{ij} | \boldsymbol{\theta}^{ij}) = \frac{\exp((\theta_k^{ij} + \mathrm{g}_k^{ij})/\tau)}{\sum_k \exp((\theta_k^{ij} + \mathrm{g}_k^{ij})/\tau)}, \; \mathrm{g}_k^{ij} \sim \mathrm{Gumbel}(0, 1). \tag{7}$$

$\mathrm{g}_k^{ij}$ is a random variable drawn from the Gumbel distribution. Note that now $\mathrm{m}_k^{ij}$ becomes a continuous random variable. It is directly differentiable with respect to $\theta_k^{ij}$ and we don't need to pass gradient through the random variable $\mathrm{g}_k^{ij}$. Therefore, the gradient of the loss function with respect to $\boldsymbol{\theta}$ can be computed as

$$\nabla_{\boldsymbol{\theta}} \mathbb{E}_{a \sim P_{\boldsymbol{\theta}}}[\mathcal{L}(\boldsymbol{m}_a, \boldsymbol{w}_a)] = \mathbb{E}_{\mathbf{g} \sim \mathrm{Gumbel}(0,1)} \left[ \frac{\partial \mathcal{L}(\boldsymbol{m}_a, \boldsymbol{w}_a)}{\partial \mathbf{m}_a} \cdot \frac{\partial \mathbf{m}_a(\boldsymbol{\theta}, \mathbf{g})}{\partial \boldsymbol{\theta}} \right]. \tag{8}$$

A temperature coefficient $\tau$ is used to control the behavior of the Gumbel Softmax. As $\tau \to \infty$, $\boldsymbol{m}^{ij}$ become continuous random variable following a uniform distribution. Therefore, in equation (3), all edges are executed and their outputs are averaged. The gradient estimation in equation (6) are biased but the variance is low, which is favorable during the initial stage of the training. As $\tau \to 0$, $\boldsymbol{m}^{ij}$ gradually becomes a discrete random variable following the categorical distribution of $P_{\boldsymbol{\theta}^{ij}}$. When computing equation (3), only one edge is sampled to be executed. The gradient estimation then becomes unbiased but the variance is high. This is favorable towards the end of the training. In our experiment, we use an exponential decaying schedule to anneal the temperature as

$$\tau = T_0 \exp(-\eta \times epoch), \tag{9}$$

where $T_0$ is the initial temperature when training begins. We decay the temperature exponentially after every epoch. Using the Gumbel Softmax trick effectively stabilizes the super net training.

In some sense, our work is in the middle ground of two previous works: ENAS by Pham et al. (2018) and DARTS by Liu et al. (2018). ENAS samples child networks from the super net to be trained independently while DARTS trains the entire super net together without decoupling child networks from the super net. By using Gumbel Softmax with an annealing temperature, our DNAS pipeline behaves more like DARTS at the beginning of the search and behaves more like ENAS at the end.

## 4.4 THE DNAS PIPELINE

Based on the analysis above, we propose a differentiable neural architecture search pipeline, summarized in Algorithm 1. We first construct a stochastic super net $G$ with architecture parameter $\boldsymbol{\theta}$ and weight $\boldsymbol{w}$. We train $G$ with respect to $\boldsymbol{w}$ and $\boldsymbol{\theta}$ separately and alternately. Training the weight $\boldsymbol{w}$ optimizes all candidate edges (operators). However, different edges can have different impact on the overall performance. Therefore, we train the architecture parameter $\boldsymbol{\theta}$, to increase the probability to sample those edges with better performance, and to suppress those with worse performance. To ensure generalization, we split the dataset for architecture search into $\mathcal{X}_{\boldsymbol{w}}$, which is used specifically to train $\boldsymbol{w}$, and $\mathcal{X}_{\boldsymbol{\theta}}$, which is used to train $\boldsymbol{\theta}$. The idea is illustrated in Fig. 1.

In each epoch, we anneal the temperature $\tau$ for gumbel softmax with the schedule in equation (9). To ensure $\boldsymbol{w}$ is sufficiently trained before updating $\boldsymbol{\theta}$, we postpone the training of $\boldsymbol{\theta}$ for $N_{warmup}$ epochs. Through the training, we draw samples $a \sim P_{\boldsymbol{\theta}}$. These sampled architectures are then trained on the training dataset $\mathcal{X}_{train}$ and evaluated on the test set $\mathcal{X}_{test}$.

---

**Algorithm 1:** The DNAS pipeline.

---

**Input:** Stochastic super net $G = (V, E)$ with parameter $\boldsymbol{\theta}$ and $\boldsymbol{w}$, searching dataset $\mathcal{X}_{\boldsymbol{w}}$ and $\mathcal{X}_{\boldsymbol{\theta}}$, training dataset $\mathcal{X}_{train}$, test dataset $\mathcal{X}_{test}$;

1   $Q_A \leftarrow \emptyset$ ;
2   **for** $epoch = 0, \cdots N$ **do**
3     $\tau \leftarrow T_0 \exp(-\eta \times epoch)$;
4     Train $G$ with respect to $\boldsymbol{w}$ for one epoch;
5     **if** $epoch > N_{warmup}$ **then**
6       Train $G$ with respect to $\boldsymbol{\theta}$ for one epoch;
7       Sample architectures $a \sim P_{\boldsymbol{\theta}}$; Push $a$ to $Q_A$;
8     **end**
9   **end**
10 **for** $a \in Q_A$ **do**
11     Train $a$ on $\mathcal{X}_{train}$ to convergence;
12     Test $a$ on $\mathcal{X}_{test}$;
13 **end**

**Output:** Trained architectures $Q_A$.

---

## 5 DNAS FOR MIXED PRECISION QUANTIZATION

We use the DNAS framework to solve the mixed precision quantization problem – deciding the optimal layer-wise precision assignment. For a ConvNet, we first construct a super net that has the same "macro-structure" (number of layers, number of filters each layer, etc.) as the given network. As shown in Fig. 2. Each node $v_i$ in the super net corresponds to the output tensor (feature map) of layer-$i$. Each candidate edge $e_k^{i,i+1}$ represents a convolution operator whose weights or activation are quantized to a lower precision.

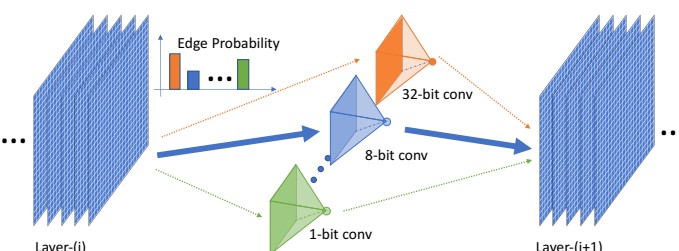

Figure 2: One layer of a super net for mixed precision quantization of a ConvNet. Nodes in the super net represent feature maps, edges represent convolution operators with different bit-widths.

In order to encourage using lower-precision weights and activations, we define the loss function as

$$\mathcal{L}(a, \boldsymbol{w}_a) = \text{CrossEntropy}(a) \times \mathcal{C}(Cost(a)). \tag{10}$$

$Cost(a)$ denotes the cost of a candidate architecture and $\mathcal{C}(\cdot)$ is a weighting function to balance the cross entropy term and the cost term. To compress the model size, we define the cost as

$$Cost(a) = \sum_{e_k^{ij} \in E} m_k^{ij} \times \#\text{PARAM}(e_k^{ij}) \times \text{weight-bit}(e_k^{ij}), \tag{11}$$

where $\#\text{PARAM}(\cdot)$ denotes the number of parameters of a convolution operator and weight-bit$(\cdot)$ denotes the bit-width of the weight. $m_k^{ij}$ is the edge selection mask described in equation (3). Alternatively, to reduce the computational cost by jointly compressing both weights and activations, we use the cost function

$$Cost(a) = \sum_{e_k^{ij} \in E} m_k^{ij} \times \#\text{FLOP}(e_k^{ij}) \times \text{weight-bit}(e_k^{ij}) \times \text{act-bit}(e_k^{ij}), \tag{12}$$

where $\#\text{FLOP}(\cdot)$ denotes the number of floating point operations of the convolution operator, weight-bit$(\cdot)$ denotes the bit-width of the weight and act-bit$(\cdot)$ denotes the bit-width of the activation. Note that in a candidate architecture, $m_k^{ij}$ have binary values $\{0, 1\}$. In the super net, we allow $m_k^{ij}$ to be continuous so we can compute the expected cost of the super net..

To balance the cost term with the cross entropy term in equation (10), we define

$$\mathcal{C}(Cost(a)) = \beta(\log(Cost(a)))^{\gamma}. \tag{13}$$

where $\beta$ is a coefficient to adjust the initial value of $\mathcal{C}(Cost(a))$ to be around 1. $\gamma$ is a coefficient to adjust the relative importance of the cost term vs. the cross-entropy term. A larger $\gamma$ leads to a stronger cost term in the loss function, which favors efficiency over accuracy.

## 6 EXPERIMENTS

### 6.1 CIFAR10 EXPERIMENTS

In the first experiment, we focus on quantizing ResNet20, ResNet56, and ResNet110 (He et al. (2016a)) on CIFAR10 (Krizhevsky & Hinton (2009)) dataset. We start by focusing on reducing model size, since smaller models require less storage and communication cost, which is important for mobile and embedded devices. We only perform quantization on weights and use full-precision activations. We conduct mixed precision search at the block level – all layers in one block use the same precision. Following the convention, we do not quantize the first and the last layer. We construct a super net whose macro architecture is exactly the same as our target network. For each block, we can choose a precision from $\{0, 1, 2, 3, 4, 8, 32\}$. If the precision is 0, we simply skip this block so the input and output are identical. If the precision is 32, we use the full-precision floating point weights. For all other precisions with $k$-bit, we quantize weights to $k$-bit fixed-point numbers. See Appendix B for more experiment details.

Our experiment result is summarized in Table 1. For each quantized model, we report its accuracy and model size compression rate compared with 32-bit full precision models. The model size is computed by equation (11). Among all the models we searched, we report the one with the highest test accuracy and the one with the highest compression rate. We compare our method with Zhu et al. (2016), where they use 2-bit (ternary) weights for all the layers of the network, except the first convolution and the last fully connect layer. From the table, we have the following observations: 1) All of our most accurate models out-perform their full-precision counterparts by up to 0.37% while still achieves 11.6 - 12.5X model size reduction. 2) Our most efficient models can achieve 16.6 - 20.3X model size compression with accuracy drop less than 0.39%. 3) Compared with Zhu et al. (2016), our model achieves up to 1.59% better accuracy. This is partially due to our improved training recipe as our full-precision model's accuracy is also higher. But it still demonstrates that our models with searched mixed precision assignment can very well preserve the accuracy.

Table 2 compares the precision assignment for the most accurate and the most efficient models for ResNet20. Note that for the most efficient model, it directly skips the 3rd block in group-1. In Fig. 3, we plot the accuracy vs. compression rate of searched architectures of ResNet110. We observe that models with random precision assignment (from epoch 0) have significantly worse compression while searched precision assignments generally have higher compression rate and accuracy.

| | | DNAS (ours) | | | TTQ (Zhu et al. (2016)) | |
|---|---|---|---|---|---|---|
| | | Full | Most Accurate | Most Efficient | Full | 2bit |
| ResNet20 | Acc | 92.35 | 92.72 (+0.37) | 92.00 (-0.35) | 91.77 | 91.13 (-0.64) |
| | Comp | 1.0 | 11.6 | 16.6 | 1.0 | 16.0 |
| ResNet56 | Acc | 94.42 | 94.57 (+0.15) | 94.12 (-0.30) | 93.20 | 93.56 (+0.36) |
| | Comp | 1.0 | 14.6 | 18.93 | 1.0 | 16.0 |
| ResNet110 | Acc | 94.78 | 95.07 (+0.29) | 94.39 (-0.39) | - | - |
| | Comp | 1.0 | 12.5 | 20.3 | - | - |

Table 1: Mixed Precision Quantization for ResNet on CIFAR10 dataset. We report results on ResNet{20, 56, 110}. In the table, we abbreviate accuracy as "Acc" and compression as "Comp".

| | g1b1 | g1b2 | g1b3 | g2b1 | g2b2 | g2b3 | g3b1 | g3b2 | g3b3 |
|---|---|---|---|---|---|---|---|---|---|
| Most Accurate | 4 | 4 | 3 | 3 | 3 | 4 | 4 | 3 | 1 |
| Most Efficient | 2 | 3 | 0 | 2 | 4 | 2 | 3 | 2 | 1 |

Table 2: Layer-wise bit-widths for the most accurate vs. the most efficient architecture of ResNet20.

## 6.2 IMAGENET EXPERIMENTS

We quantize ResNet18 and ResNet34 on the ImageNet ILSVRC2012 (Deng et al. (2009)) dataset. Different from the original ResNet (He et al. (2016a)), we use the "ReLU-only preactivation" ResNet from He et al. (2016b). Similar to the CIFAR10 experiments, we conduct mixed precision search at the block level. We do not quanitze the first and the last layer. See Appendix B for more details.

We conduct two sets of experiments. In the first set, we aim at compressing the model size, so we only quantize weights and use the cost function from equation (11). Each block contains convolution operators with weights quantized to $\{1, 2, 4, 8, 32\}$-bit. In the second set, we aim at compressing computational cost. So we quantize both weights and activations and use the cost function from equation (12). Each block in the super net contains convolution operators with weights and activations quantized to $\{(1, 4), (2, 4), (3, 3), (4, 4), (8, 8), (32, 32)\}$-bit. The first number in the tuple denotes the weight precision and the second denotes the activation precision. The DNAS search is very efficient, taking less than 5 hours on 8 V100 GPUs to finish the search on ResNet18.

Our model size compression experiment is reported in Table 3. We report two searched results for each model. "MA" denotes the searched architecture with the highest accuracy, and "ME" denotes the most efficient. We compare our results with TTQ (Zhu et al. (2016)) and ADMM (Leng et al. (2017)). TTQ uses ternary weights (stored by 2 bits) to quantize a network. For ADMM, we cite the result with $\{-4, 4\}$ configuration where weights can have 7 values and are stored by 3 bits. We report the accuracy and model size compression rate of each model. From Table 3, we have the following observations: 1) All of our most accurate models out-perform full-precision models by up to 0.5% while achieving 10.6-11.2X reduction of model size. 2) Our most efficient models can achieve 19.0 to 21.1X reduction of model size, still preserving competitive accuracy. 3) Compared with previous works, even our less accurate model has almost the same accuracy as the full-precision model with 21.1X smaller model size. This is partially because we use label-refinery (Bagherinezhad et al. (2018)) to effectively boost the accuracy of quantized models. But it still demonstrate that our searched models can very well preserve the accuracy, despite its high compression rate.

| | | DNAS (ours) | | | TTQ | ADMM |
|---|---|---|---|---|---|---|
| | | Full | MA | ME | Full | 2bit | 3bit |
| ResNet18 | Acc | 71.03 | 71.21 (+0.18 ) | 69.58 (-1.45) | 69.6 | 66.6 (-3.0) | 68.0 (-1.6) |
| | Comp | 1.0 | 11.2 | 21.1 | 1.0 | 16.0 | 10.7 |
| ResNet34 | Acc | 74.12 | 74.61 (+0.49) | 73.37 (-0.75) | - | | |
| | Comp | 1.0 | 10.6 | 19.0 | | | |

Table 3: Mixed Precision Quantization for ResNet on ImageNet for model size compression. In the table, we abbreviate accuracy as "Acc" and compression as "Comp". "MA" denotes the most accurate model from architecture search and "ME" denotes the most efficient model.

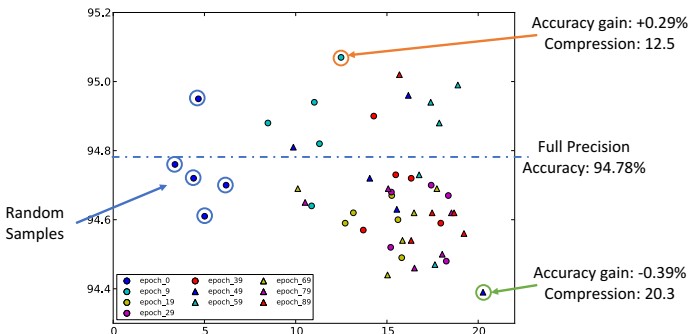

Figure 3: Visualization of all searched architectures for ResNet110 and CIFAR10 dataset. x-axis is the compression rate of each model. y-axis is the accuracy.

| | | DNAS (ours) | | | PACT | DoReFA | QIP | GroupNet |
|---|---|---|---|---|---|---|---|---|
| | | arch-1 | arch-2 | arch-3 | W4A4 | W4A4 | W4A4 | W1A2G5 |
| ResNet18 | Acc | 71.01 | 70.64 | 68.65 | 69.2 | 68.1 | 69.3 | 67.6 |
| | Full Acc | 71.03 | 71.03 | 71.03 | 70.2 | 70.2 | 69.2 | 69.7 |
| | Acc $\Delta$ | -0.02 | -0.39 | -2.38 | -1.0 | -2.1 | +0.1 | -2.1 |
| | Comp | 33.2 | 62.9 | 103.5 | 64 | 64 | 64 | 102.4 |
| ResNet34 | Acc | 74.21 | 73.98 | 73.23 | | | | |
| | Full Acc | 74.12 | 74.12 | 74.12 | | | - | |
| | Acc $\Delta$ | +0.09 | -0.14 | -0.89 | | | | |
| | Comp | 40.8 | 59.0 | 87.4 | | | | |

Table 4: Mixed Precision Quantization for ResNet on ImageNet for computational cost compression. We abbreviate accuracy as "Acc" and compression rate as "Comp". "arch-{1, 2, 3}" are three searched architectures ranked by accuracy.

Our experiment on computational cost compression is reported in Table 4. We report three searched architectures for each model. We report the accuracy and the compression rate of the computational cost of each architecture. We compute the computational cost of each model using equation (12). We compare our results with PACT (Choi et al. (2018)), DoReFA (Zhou et al. (2016)), QIP (Jung et al. (2018)), and GroupNet (Zhuang et al. (2018)). The first three use 4-bit weights and activations. We compute their compression rate as $(32/4) \times (32/4) = 64$. GroupNet uses binary weights and 2-bit activations, but its blocks contain 5 parallel branches. We compute its compression rate as $(32/1) \times (32/2)/5 \approx 102.4$ The DoReFA result is cited from Choi et al. (2018). From table 4, we have the following observations: 1) Our most accurate architectures (arch-1) have almost the same accuracy (-0.02% or +0.09%) as the full-precision models with compression rates of 33.2x and 40.8X. 2) Comparing arch-2 with PACT, DoReFa, and QIP, we have a similar compression rate (62.9 vs 64), but the accuracy is 0.71-1.91% higher. 3) Comparing arch-3 with GroupNet, we have slightly higher compression rate (103.5 vs. 102.4), but 1.05% higher accuracy.

# 7 CONCLUSION

In this work we focus on the problem of mixed precision quantization of a ConvNet to determine its layer-wise bit-widths. We formulate this problem as a neural architecture search (NAS) problem and propose a novel, efficient, and effective differentiable neural architecture search (DNAS) framework to solve it. Under the DNAS framework, we efficiently explore the exponential search space of the NAS problem through gradient based optimization (SGD). We use DNAS to search for layer-wise precision assignment for ResNet on CIFAR10 and ImageNet. Our quantized models with 21.1x smaller model size or 103.9x smaller computational cost can still outperform baseline quantized or even full precision models. DNAS is very efficient, taking less than 5 hours to finish a search on ResNet18 for ImageNet. It is also a general architecture search framework that is not limited to the mixed precision quantization problem. Its other applications will be discussed in future publications.

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

## APPENDIX A  WEIGHT AND ACTIVATION QUANTIZATION

For readers' convenience, we describe the functions we use to quantize weights and activations in this section. We follow DoReFa-Net (Zhou et al. (2016)) to quantize weights as

$$w_k = 2Q_k(\frac{\tanh(w)}{2\max(|\tanh(w)|)} + 0.5). \qquad (14)$$

$w$ denotes the latent full-precision weight of a network. $Q_k(\cdot)$ denotes a $k$-bit quantization function that quantizes a continuous value $w \in [0, 1]$ to its nearest neighbor in $\{\frac{i}{2^k-1}|i = 0, \cdots, 2^k - 1\}$. To quantize activations, we follow Choi et al. (2018) to use a bounded activation function followed by a quantization function as

$$y = PACT(x) = 0.5(|x| - |x - \alpha| + \alpha),$$
$$y_k = Q_k(y/\alpha) \cdot \alpha. \qquad (15)$$

Here, $x$ is the full precision activation, $y_k$ is the quantized activation. $PACT(\cdot)$ is a function that bounds the output between $[0, \alpha]$. $\alpha$ is a learnable upper bound of the activation function.

## APPENDIX B  EXPERIMENT DETAILS

We discuss the experiment details for the CIFAR10 experiments. CIFAR10 contains 50,000 training images and 10,000 testing images to be classified into 10 categories. Image size is $32 \times 32$. We report the accuracy on the test set. To train the super net, we randomly split 80% of the CIFAR10 training set to train the weights $w$, and 20% to train the architecture parameter $\theta$. We train the super net for 90 epochs with a batch size of 512. To train the model weights, we use SGD with momentum with an initial learning rate of 0.2, momentum of 0.9 and weight decay of $5 \times 10^{-4}$. We use the cosine decay schedule to reduce the learning rate. For architecture parameters, we use Adam optimizer (Kingma & Ba (2014)) with a learning rate of $5 \times 10^{-3}$ and weight decay of $10^{-3}$. We use the cost function from equation (11). We set $\beta$ from equation (13) to 0.1 and $\gamma$ to 0.9. To control Gumbel Softmax functions, we use an initial temperature of $T_0 = 5.0$, and we set the decaying factor $\eta$ from equation (9) to be 0.025. After every 10 epochs of training of super net, we sample 5 architectures from the distribution $P_\theta$. We train each sampled architecture for 160 epochs and use cutout (DeVries & Taylor (2017)) in data augmentation. Other hyper parameters are the same as training the super net.

We next discuss the experiment details for ImageNet experiments. ImageNet contains 1,000 classes, with roughly 1.3M training images and 50K validation images. Images are scaled such that their shorter side is 256 pixels and are cropped to $224 \times 224$ before feeding into the network. We report the accuracy on the validation set. Training a super net on ImageNet can be very computationally expensive. Instead, we randomly sample 40 categories from the ImageNet training set to train the

super net. We use SGD with momentum to train the super net weights for 60 epochs with a batch size of 256 for ResNet18 and 128 for ResNet34. We set the initial learning rate to be 0.1 and reduce it with the cosine decay schedule. We set the momentum to 0.9. For architecture parameters, we use Adam optimizer with the a learning rate of $10^{-3}$ and a weight decay of $5 \times 10^{-4}$. We set the cost coefficient $\beta$ to 0.05, cost exponent $\gamma$ to 1.2. We set $T_0$ to be 5.0 and decay factor $\eta$ to be 0.065. We postpone the training of the architecture parameters by 10 epochs. We sample 2 architectures from the architecture distribution $P_{\boldsymbol{\theta}}$ every 10 epochs. The rest of the hyper parameters are the same as the CIFAR10 experiments. We train sampled architectures for 120 epochs using SGD with an initial learning rate of 0.1 and cosine decay schedule. We use label-refinery (Bagherinezhad et al. (2018)) in training and we use the same data augmentation as this Pytorch example[1].

---

[1] https://github.com/pytorch/examples/tree/master/imagenet

