# OpenReview forum: "Mixed Precision Quantization of ConvNets via Differentiable Neural Architecture Search"
_ICLR.cc/2019/Conference_

### Official Review · AnonReviewer2 · 2018-11-01
**An interesting topic with promising experiment results.**

**Rating:** 6
**Confidence:** 3

**Review:**

This paper presents a new approach in network quantization. The key insights of this paper is quantizing different layers with different bit-widths, instead of using fixed 32-bit width for all layer weights and activation in previous works. At the same time, this paper adopted the idea form both DARTS and ENAS with parameter sharing, and introduces a new differentiable neural architecture search framework. As the authors proposed, this DNAS framework is able to search efficiently and effective through a large search space.  As demonstrated in the Experiment section of the paper, it achieves better validation accuracy than ResNet with much smaller model size and lower computational cost.

1. An improved gradient method in updating the network architecture and parameters compared to DARTS and ENAS. It applies the Gumbel softmax to refine the sub-graph structure without training the entire super-net through the whole process. The work is able to obtain the same level of validation accuracy on Cifar-10 as ResNet while reduce the model parameters by a large margin.
2. The work is in the middle ground of two previous works: ENAS by Pham et al. (2018) and DARTS by Liu et al. (2018). However, there is no comparison with ENAS and DARTS in experiments. ENAS samples child networks from the super net to be trained independently while DARTS trains the entire super net together without decoupling child networks from the super net. By using Gumbel Softmax with an annealing temperature, The proposed DNAS pipeline behaves more like DARTS at the beginning of the search and behaves more like ENAS at the end.

---

> ### Author Response · Authors · 2018-11-06
> **Thank you for your review**
>
> Thank you for your review.
>
> Your summary correctly and comprehensively reflects the gist of our paper. One minor correction we would like to make is that our experiments are not only conducted on the Cifar-10 dataset. On ImageNet dataset, we were able to compress ResNet models with no or little accuracy loss, but reduce the model size by up to 21.1x and computational cost by up to 103.5x, better than previous baselines.
>
> Please let us know if you have further questions or concerns that we can help clarify.

---

### Official Review · AnonReviewer1 · 2018-11-07
**Neural Architecture Search Approach to Network Quantization**

**Rating:** 7
**Confidence:** 3

**Review:**

In this work the authors introduce a new method for neural architecture search (NAS) and use it in the context of network compression. Specifically, the NAS method is used to select the precision quantization of the weights at each layer of the neural network. Briefly, this is done by first defining a super network, which is a DAG where for each pair of nodes, the output node is the linear combination of the outputs of all possible operations (i.e., layers with different precision quantizations). Following [1], the weights of the linear combination are regarded as the probabilities of having certain operations (i.e., precision quantization), which allows for learning a probability distribution over the considered operations. Differently from [1], however, the authors bridge the soft sampling in [1] (where all operations are considered together but weighted accordingly to the corresponding probabilities) to a hard sampling (where a single operation is considered with the corresponding probability) through an annealing procedure based on the Gumbel Softmax technique. Through the proposed NAS algorithm, one can learn a probability distribution on the operations by minimizing a loss that accounts for both accuracy and model size. The final output of this search phase is a set of sampled architectures (containing a single operation at each connection between nodes), which are then retrained from scratch. In applications to CIFAR-10 and ImageNet, the authors achieve (and sometime surpass) state-of-the-art performance in model compression.

The two contributions of this work are
1)	A new approach to weight quantization using principles of NAS that is novel and promising;
2)	New insights/technical improvements in the broader field of NAS. While the utility of the method in the more general context of NAS has not been shown, this work will likely be of interest to the NAS community.

I only have one major concern. The architectures are sampled from the learnt probability distribution every certain number of epochs while training the supernet. Why? If we are learning the distribution, would not it make sense to sample all architectures only after training the supernet at our best?
This reasoning leads me to a second question. In the CIFAR-10 experiments, the authors sample 5 architecture every 10 epochs, which means 45 architectures (90 epochs were considered). This is a lot of architectures, which makes me wonder: how would a “cost-aware” random sampling perform with the same number of sampled architectures?

Also, I have some more questions/minor concerns:

1)	The authors say that the expectation of the loss function is not directly differentiable with respect to the architecture parameters because of the discrete random variable. For this reason, they introduce a Gumbel Softmax technique, which makes the mask soft, and thus the loss becomes differentiable with respect to the architecture parameters. However, subsequently in the manuscript, they write that Eq 6 provides an unbiased estimate for the gradients. Do they here refer to the gradients with respect to the weights ONLY? Could we say that the advantage of the Gumbel Softmax technique is two-fold? i) make the loss differentiable with respect to the arch parameters; ii) reduce the variance of the estimate of the loss gradients with respect to the network weights.

2)	Can the author discuss why the soft sampling procedure in [1] is not enough? I have an intuitive understanding of this, but I think this should be clearly discussed in the manuscript as this is a central aspect of the paper.

3)	The authors use a certain number of warmup steps to train the network weights without updating the architecture parameters to ensure that “the weights are sufficiently trained”. Can the authors discuss the choice on the number of warmup epochs?

I gave this paper a 5, but I am overall supportive. Happy to change my score if the authors can address my major concern.

[1] Liu H, Simonyan K, Yang Y. Darts: Differentiable architecture search. arXiv preprint arXiv:1806.09055. 2018 Jun 24.

-----------------------------------------------------------
Post-Rebuttal
---------------------------------------------------------
The authors have fully addressed my concerns. I changed the rating to a 7.

---

> ### Author Response · Authors · 2018-11-22
> **Thank you for your review.**
>
> We want to thank the reviewer#1 for your feedback. Your summary correctly reflects the content of our paper. We hope this rebuttal can address your concerns.
>
> Major concern: Trained sampling vs random sampling
> We sample architectures every a few epochs, mainly because in our experiments, we want to analyze the behavior of the architecture distribution at different super net training epochs. This analysis is illustrated in figure 3 of our paper. We can see that at epoch-0, where the architecture distribution is trained for only one epoch (close to random sampling), the sampled architectures have much lower compression rate. Similarly, for epoch-9, architectures also have relatively low compression rate. In comparison, at epoch-79 and epoch-89, architectures have higher compression rates and accuracy. The difference between epoch-79 vs. epoch-89 is small since the distribution has converged.
>
> As the reviewer#2 suggests, we can train the super net until the last epoch, then sample and train architectures from this distribution. Figure 3 shows that the five architectures sampled at epoch-89 are much better than the five architectures at epoch-0, which are essentially drawn from random sampling. Also, note that for CIFAR10-ResNet-110 experiments, the search space contains 7^54 = 4x10^45 possible architectures, 45 sampled architectures are tiny compared with the search space.
>
> Reviewer #2 suggests comparing with a “cost-aware” random sampling policy. We tried a simple baseline that at each layer, we sample a conv operator with b-bit precision with probability
>         prob(precision=b) ~  1/(1 + b)
> The performance of this policy is much worse since for a conv operator with precision-0 (in our notation, bit-0 denotes we skip the layer), the sampling probability is 33x higher than full-precision convolution, 2x higher than 1-bit, 3x higher than 2-bit, and so on. Architectures sampled from this distribution are extremely small but with much worse accuracy. We understand this might not be the best “cost-aware” sampling policy. If reviewer#1 has better suggestions, we are happy to try.
>
> Minor concern #1: Value of the Gumbel Softmax function
> Yes. We agree with the comments that the advantages of the Gumbel Softmax technique are two-fold:
>    1. It makes the loss function differentiable with respect to the architecture parameter \theta.
>    2. Compared with other gradient estimation techniques such as Reinforce, Gumbel Softmax balances the variance/bias of the gradient estimation with respects to weights.
>
> Minor concern #2: Comparison with non-stochastic method such as DARTS
> DARTS [1] does not really sample candidate operators during the forward pass. Outputs of candidate operators are multiplied with some coefficients and summed together. For the problem of mixed precision quantization, this can be problematic. Let's consider a simplified scenario
>                           y = alpha_1 * y_1 + alpha_2 * y_2
> Let's assume both y_1 and y_2 are in binary and are in {0, 1}. Assuming alpha_1=0.5 and alpha_2=0.25, then the possible values of y are {0, 0.25, 0.5, 0.75}, which essentially extend the effective bit-width to 2 bit. This is good for the super net's accuracy, but the performance of the super net cannot transfer to the searched architectures in which we have to pick only one operator per layer. Using our method, however, the sampling ensures that the super net only picks one operator at a time and the behavior can transfer to the searched architectures.
>
> Minor concern #3: Warmup training
> We use warmup training since in our ImageNet experiments. We observe that at the beginning of the super net training, the operators are not sufficiently trained, and their contributions to the overall accuracy are not clear, but their cost differences are always significant. As a result, the search always picks low-cost operators. To prevent this, we use warmup training to ensure all the candidate operators are sufficiently trained before we optimize architecture parameters. In our ImageNet experiments, we found that ten warmup epochs are good enough. In CIFAR-10 experiments, warmup training is not needed.

---

### Official Review · AnonReviewer4 · 2018-11-12
**Inetresting approach to quantization and interesting experimental results**

**Rating:** 6
**Confidence:** 3

**Review:**

The authors propose a network quantization approach with adaptive per layer bit-width. The approach is based on a network architecture search (NAS) method. The authors aim to solve the NAS problem through SGD. Therefore, they propose to first reprametrize the the discrete random variable determining if an edge is computed or not to make it differentiable and then use Gumbel Softmax function as a way to effectively control the variance of the obtained unbiased estimator. This variance can indeed make the convergence of the procedure hard. The procedure is then adapted to the problem of network quantization with different band-widths.

The proposed approach is interesting. The differerentiable NAS procedure is particularly important and can have an important impact. The idea of having an adaptive per layer precision is also well motivated, and shows competitive (if not better) results empirically.

Some additional experiments can make the paper stronger:
* Compare the result of the procedure to an exhaustive search in a setting where the latter is feasible (shallow architecture on an easy task with few possible bit widths)
* Compare the procedure to other state of the art NAS procedures (DARTS and ENAS) with the same search space adapted to the quantization problem, to empirically show that the proposed procedure is a compromise between these two methods as claimed by the authors.

---

> ### Author Response · Authors · 2018-11-22
> **Reply to suggestions on more experiments**
>
> We want to thank reviewer#4 for your review. Your summary correctly reflects the content of our paper.
>
> We want to comment on the suggestions for new experiments:
>
> Comparing with exhaustive search: This is a good idea. However, one concern is that since the search space is combinatorial, even a shallow network (e.g., 5) with a smaller number of precisions (e.g., 32, 8, 1) can have a large search space of (e.g., 3^5 = 243 architectures) for which exhaustive search is intractable.
>
> Comparing with DARTS and ENAS: ENAS is not open-sourced, so a direct comparison is difficult. A more detailed analysis comparing DNAS with DARTS is discussed in the reply to reviewer#1, minor concern #2: https://openreview.net/forum?id=BJGVX3CqYm&noteId=S1lyG-h7A7
>
> We plan to perform the suggested experiments of comparing with exhaustive search and DARTS. The results will be hopefully updated before the revision deadline and the camera-ready if the paper is accepted.

---

### Official Review · AnonReviewer3 · 2018-11-12
**Contribution not significant; Potentially covered by prior work**

**Rating:** 5
**Confidence:** 5

**Review:**

The paper approaches the bit quantization problem from the perspective of neural architecture search, by treating each possible precision as a different type of neural layer. They estimate the proportion of each layer using a gumbel-softmax reparametrization. Training updates parameters and these proportions alternately.

The authors claim that prior work has only dealt with uniform bit precision. This is clearly false e.g.
https://arxiv.org/pdf/1807.00942.pdf
https://arxiv.org/abs/1708.04788
https://arxiv.org/pdf/1705.08665.pdf

In particular, https://arxiv.org/pdf/1807.00942.pdf uses the same approach, using gumbel-softmax to estimate the best number of bits. In the least, the authors needs to mention and contrast their approach, e.g. they can handle a budget constraint, but they use a fixed quantization function.

There is an inherent strength in this approach that the authors have not fully explored. The most recent key discovery in low precision networks is that the optimal parameters take very different values depending on the precision, ie beyond simple clipping/snapping based on quantization error. The DNAS approach can capture this, because the parameters of different precisions need not be constrained via a fixed quantization/activation function (appendix B). Therefore the following questions become important to understand.

1. How are the weights w updated for low precision. I understand that you first sample an architecture but there is no explanation of how the low bit (e.g. 1-bit) weights are updated. Do you update the 32-bit weights, then use the functions in Appendix B to derive the low bit parameters? This is much less interesting than the power of the DNAS idea. Do you directly update them using STE?
2. Why is it important to train in an alternating fashion? How did you split the training set in to two for each ? Why not use a single training set?
3. Are the "edge probabilities" over different precision in any way the function of the input (image)? It seems your approach is able to distinguish "easy" and "hard" images by increasing the precision of parameters. If so, this should be explained and demonstrated.
4. In Eq (10), it is unusual to take the product of network performance and penalty term for parsimony. This needs to be explained vs. taking a sum of the two terms which has the nice interpretation of being the lagrangian of a constrained optimization problem. Do you treat these as instance level weights?
5. Experiments only show ResNet architecture, whereas prior work showed a broaded set of results. Only TTQ and ADMM is compared, where the most relevant work is https://arxiv.org/pdf/1807.00942.pdf. It is not clear if the good performance comes due to the block connectivity structure with skip connections, combined with the fact that the first and last layers are not quantized.

---

> ### Author Response · Authors · 2018-11-22
> **Thank you for your review, but your understanding of our paper or the previous work is wrong.**
>
> First, we would like to thank the reviewer for pointing to previous works on mixed precision quantization. We were not aware of them and are happy to acknowledge these prior works in our paper. However, we strongly disagree with the reviewer's opinion that our method is the same as, or covered by [1].
>
> [1] introduces an interesting technique that uses Gumbel Softmax to determine the precision allocation for each layer of a neural network. It proposes a precision allocation process to assign each bit from a "bit pool" to different layers of a network. For each bit, it uses Gumbel Softmax to compute a "soft-allocation" to determine where the bit is assigned to.  The number of bit for a layer is the sum of all the bits assigned to the layer. It modifies the quantization function to allow non-integer bit quantization and uses STE to compute the gradient of the bit allocation.
>
> Our approach is fundamentally different from [1] in the following aspects:
> 1. Problem formulation and scope: We formulate the problem in a more general way that we support arbitrary layer-wise operator selection. Under our framework, the operator can be convolution with different precisions or any other operators such as max pooling. So our method can be applied to more general neural architecture search problems. In comparison, [1] only works for mixed precision quantization since the formulation of [1] is to assign a pool of bits to layers of a network.
> 2. Algorithm procedure: we conduct architecture search by training a stochastic super net to determine the layer-wise operator type. In comparison, [1] starts with a pool of bits, and assign each bit from the pool to a layer.
> 3. Gumbel Softmax: Our method lets each layer choose a different precision. The Gumbel Softmax function controls a probability that for each layer, which operator (precision) to choose. [1] allocates a bit to a different layer, and the Gumbel softmax function determines which layer should the bit be assigned to.
> 4. Performance: Each candidate operator in our super net has independent weights and activations. Our method allows the weights and activations for different precisions to have different "latent" full-precision values, which is the key to a good quantization performance. In [1] however, each layer has only one weight/activation and is quantized to different precisions as the training proceed.  As also mentioned by the reviewer, directly mapping a higher precision weights/activations to lower precisions can lead to performance degradation.
>
> Given such obvious and fundamental differences, we cannot agree that [1] covered the technical contribution of our paper.

---

> > ### Author Response · Authors · 2018-11-22
> > **Continued discussion**
> >
> > To address the reviewer's additional questions:
> > Question 1: How are weights updated
> > Following [2,3], we use equation (14, 15) in appendix A to quantize each candidate operators' weight and activations for both super net and searched architectures. We do not treat ultra-low precision weights and activations differently. The gradient update w.r.t. full-precision weights and activations are well described in [2, 3], and we use the same approach.
> >
> > Question 2: Why alternatively train model weights and architecture parameters
> > This ensures the operator choices do not overfit the training set and can be generalized to the validation set. This is a widely adopted technique in neural architecture search literature such as [4,5]. As we described in Appendix B, we randomly sample 80% of the training set to train the weights and 20% to train the architecture parameters.
> >
> > Question 3: Is the edge probability conditioned on the input
> > No, the edge probability is not conditioned on the input. Although this is an interesting idea (dynamic neural networks conditioned on the input), it is not the scope of this paper.
> >
> > Question 4: Why loss function multiply instead of sum two components
> > Our neural architecture search problem can be seen as a multi-objective optimization problem, and we use the weighted product model to construct a loss function by multiplying the cross-entropy term with the log-cost term. This is also used in [6,7]. In our experiments, we tried summing or multiplying the two terms, and we found that multiplying works better.
> >
> > Question 5: Why not compare [1]'s result on AlexNet
> > [1]'s experiment is on AlexNet. In the NN quantization research, AlexNet is known to be redundant, and many methods can drastically quantize AlexNet without accuracy loss. Specifically, SqueezeNet [8] shows it can reduce the model size of AlexNet by 500x without accuracy loss. Therefore, we do not think quantizing AlexNet is still a good benchmark to show the effectiveness of new quantization methods. TTQ [9] shows it can quantize AlexNet weights to 2 bit (ternary) without accuracy loss, but on ResNet18, TTQ shows 3% accuracy loss, proving that quantizing ResNet is more difficult. When quantizing both activations and weights, [1]'s best result on AlexNet is 52.54% top-1 accuracy and each layer on average can have 8 bits. This accuracy is worse than the 2-bit quantization of DoReFa-Net[2] (53.6%) and PACT[3] (55.0%). As a result, comparing our method with DoReFa-Net and PACT and showing better performance is sufficient to prove the effectiveness of our method.
> >
> > [1] https://arxiv.org/pdf/1807.00942.pdf
> > [2] https://arxiv.org/abs/1606.06160
> > [3] https://arxiv.org/abs/1805.06085
> > [4] https://arxiv.org/abs/1806.09055
> > [5] https://arxiv.org/abs/1802.03268
> > [6] https://arxiv.org/pdf/1802.03494.pdf
> > [7] https://arxiv.org/pdf/1807.11626.pdf
> > [8] https://arxiv.org/abs/1602.07360
> > [9] https://arxiv.org/abs/1612.01064

---

> > > ### Comment · AnonReviewer3 · 2018-11-30
> > > **reply to response**
> > >
> > > Q4 is still unsatisfactory to me. I would at least like to see the two results in comparison. It seems you are scaling the gradient of one term with the second term.
> > >
> > > Based on the feedback, and the existing experimental deficiencies, and the remaining questions above, I am happy to increase my score.

---

> > > > ### Author Response · Authors · 2018-12-01
> > > > **Response**
> > > >
> > > > | Q4 is still unsatisfactory to me. I would at least like to see the two results in comparison. It seems you are scaling the gradient of one term with the second term.
> > > >
> > > > We are happy to add some ablation studies on this in the appendix once we can update the paper. However, I think this is a minor point -- using summation or multiplication in the loss function does not make a big difference to the technical contribution of this paper.

---

> > ### Comment · AnonReviewer3 · 2018-11-30
> > **reply to response**
> >
> > The point about "bit assigned to layer" vs "assigning precision per layer" is a minor one, but I appreciate that the approach is slightly different.
> >
> > Re 4. In your paper, this is not clear AT ALL and I think it is a very important point. I had asked this question in my original review as well. How are the latent 32-bit weights handled? How do you do gradient descent e.g. on the corresponding latent weights? Are latent 32-bit weights even mentioned in the paper? I had to go to the appendix to find the quantization function. This needs to be addressed.

---

> > > ### Author Response · Authors · 2018-12-01
> > > **Response**
> > >
> > > Thank you for your reply.
> > >
> > > | The point about "bit assigned to layer" vs "assigning precision per layer" is a minor one, but I appreciate that the approach is slightly different.
> > >
> > > We believe that our approach is *fundamentally*, instead of slightly, different from [1]. In the previous response, we have listed four significant differences. At the top level, [1] proposes a smart quantization function that supports quantizing full-precision numbers to NON-INTEGER precisions. This allows [1] to "softly" assign each bit from a pool of bits to different layers with fixed operators. Our method, however, focuses on determining layer-wise operators, and the operators happen to be convolutions with different INTEGER precisions, but they can also be other types of operators such as max pooling. Our formulation is more general, therefore our method can be applied to general neural architecture search problems.
> > >
> > > | Re 4. In your paper, this is not clear AT ALL and I think it is a very important point. I had asked this question in my original review as well. How are the latent 32-bit weights handled? How do you do gradient descent e.g. on the corresponding latent weights? Are latent 32-bit weights even mentioned in the paper? I had to go to the appendix to find the quantization function. This needs to be addressed.
> > >
> > > This is a good advice. we hope point-4 in the previous response clarified your question: at a given layer, each candidate operator has independent full-precision latent weight and activation. Latent weights are quantized following DoReFa-Net[2] and activations are quantized following PACT [3]. Due to the page limit, we moved this part to the appendix since we quantize weights and activations using the existing methods [2, 3]. We will add more explanations to this part to make it clearer.

---

### Public Comment · (anonymous) · 2018-12-06
**The compression rate metric is wrong, so the experimental results are not comparable.**

Dear authors and reviewers,

This paper proposes to use an improved version of DARTs for searching the precision for each convolutional layer. Since different layers have different precisions, the complexity calculation is extremely important for fair comparision.
But I find the compression rate metric for $k$-bit you use in this paper is simply $32 / k$. However, this does not make sense.
According to the bitwise operations for fixed-point approaches[1, 2], the compression ratio of $k$-bit quantization is proportional to $2^k$ rather than  $k$. For example, the compression rate reduction of  2-bit compared to 4-bit should be $2^4 / 2^2 = 4$ rather than $4 / 2 = 2$.  I think this is a big issue and you really need to clarify this, otherwise the experimental results are not comparable at all.

[1]: https://arxiv.org/abs/1606.06160
[2]:http://openaccess.thecvf.com/content_ECCV_2018/papers/Dongqing_Zhang_Optimized_Quantization_for_ECCV_2018_paper.pdf

---

> ### Author Response · Authors · 2018-12-06
> **Thank you for your question, but the computational cost is not exponential with respect to the bit-width.**
>
> Thank you for your question. In our paper, we conduct mixed precision quantization in two different settings: model size compression and computational cost reduction.
>
> Model size compression: In this setting, our goal is to reduce the model size (parameter size * bit-width), and we only quantize weights. I think there's no doubt here that the compression rate can be directly computed by 32/k.
>
> Computational cost reduction: In this setting, we quantize both weights and activations. Depending on the implementation details, if we adopt the bit-wise operation by equation (3, 4) from [1] (DoReFaNet), the computational cost is proportional to (k_w*k_a), where k_w is the weight's bit-width, and k_a is the activation's bit-width. This is exactly how we compute the computational cost (and compression rate) in our paper.
>
> In addition, for all the baseline methods in our paper, we compute the reduction rate in the same manner to make sure that our results are consistent and comparable.
>
> Hope this clarifies your question.
>
> [1]: https://arxiv.org/abs/1606.06160

---

> > ### Public Comment · (anonymous) · 2018-12-06
> > **Better, but still unclear to me.**
> >
> > Dear authors,
> >
> > Thanks for your response.  Assume we only consider the model size compression here. And I am still a little confusing with respect to "in this setting, our goal is to reduce the model size (parameter size * bit-width)". According to DOREFA-NET, during inference, the value range for 4-bit weights should be [-2^4, 2^4] (only assume uniform quantization, and use XNOR bitwise operation.). Similarly, the representation range for 2-bit weights is [-2^2, 2^2].  So model size = parameter size * bit-width is not accurate.
> >
> > And you need to explain that the compression rate means computational cost in the paper clearly in order to avoid misunderstanding, since model size compression and computational cost reduction are two different things.

---

> > > ### Public Comment · (anonymous) · 2018-12-06
> > > **Agree with anonymous**
> > >
> > > I agree with anonymous, you can check this ICML 2017 paper where computational and representational costs are defined. One is a quadratic function of precision while the other is a linear function of precision. Perhaps the authors should use this reference to clarify matters.
> > >
> > > Analytical Guarantees on Numerical Precision of Deep Neural Networks - by Sakr et al. http://proceedings.mlr.press/v70/sakr17a.html

---

> > > > ### Author Response · Authors · 2018-12-06
> > > > **Thank you for your suggestion.**
> > > >
> > > > I agree with your post that the "computational cost" is quadratic (if the activation and weights are quantized to the same precision) and the representational cost is linear. But the anonymous-1's post claims the cost is exponential. And your post said "I agree with anonymous", which actually contradicts your claim. Can you clarify?
> > > >
> > > > I think it is a norm that the term "model size" really just mean "representational cost" or the storage space of an NN model. We also defined the "computational cost" in the paper.
> > > >
> > > > However, I think your suggestion is very good and I will update the paper to better clarify these basic definitions.

---

> > > > > ### Public Comment · (anonymous) · 2018-12-06
> > > > > **Clarification about agreement**
> > > > >
> > > > > I only meant I agree that the computational cost is not a linear function of precision. Sorry about the confusion and thanks for the reply.

---

> > > ### Author Response · Authors · 2018-12-06
> > > **Value range != model size (storage space)**
> > >
> > > As the title suggests, value range is not equivalent to model size.
> > >
> > > By model size, we mean the storage space for a model. It is simply computed by #parameters x bit-width. It is true that the value range for an N-bit weight is [0, 2^N-1], but it still takes N-bit to store the weight and therefore, the model size (storage space) of a layer is computed by #params x bit-width.
> > >
> > > > And you need to explain that the compression rate means computational cost in the paper clearly in order to avoid misunderstanding, since model size compression and computational cost reduction are two different things.
> > >
> > > Yes, we put two experiments (on for model size, another for computational cost) results in two tables. In the experiment section and the cost function definition, we also explain the differences. We are happy to adopt suggestions how we can make this clearer.
> > >
> > > Thanks.

---

> > > > ### Public Comment · (anonymous) · 2018-12-06
> > > > **Different understanding of model size**
> > > >
> > > > Dear authors,
> > > >
> > > > Thanks for your kindly reply. I think the problem locates at the different understanding of model size. My answer is based on the "on the fly" model size, which means how much memory and accumulator bandwidth you need during the inference stage.  However, your answer is based on the "offline" storage definition, which means how much memory you need for storaging the model on a hard disk. But I think the "on the fly" memory consumption is more meaningful and the issue becomes clear now.

---

> > > > > ### Author Response · Authors · 2018-12-06
> > > > > **On-the-fly model size**
> > > > >
> > > > > Glad to see that things got clarified. However, I don't quite understand what you mean by "on-the-fly" model size and why it's exponential w.r.t. precisions. Could you clarify?

---

> > > > > > ### Public Comment · (anonymous) · 2018-12-06
> > > > > > **Clarification with respect to "On-the-fly" and "exponential w.r.t. precisions"**
> > > > > >
> > > > > > Dear authors,
> > > > > >
> > > > > > 1: "On-the-fly" means the RUN-TIME memory consumption during the inference stage, where you need to consider the numerical precision. For example, you have 2-bit weights and 2-bit activations, a possible choice is in range {0,1,2,4}. In this 2-bit multiplication, when both numbers are 4, it outputs 4 × 4 = 16, which is not within the range.
> > > > > > "Offline" means the memory space you need for saving the model on the hardware, which corresponds to the definition in the paper.  The memory consumption for these two cases are different.
> > > > > >
> > > > > > 2: Actually, I just want to claim that  using the "linear" metric in your paper is "definitely" not appropriate. Of course, my "exponential" claim is also not accurate. According to another anonymous, he provided a better metric [1].
> > > > > > And I also recommend another bitwise metric (BOPs) which can be referred to [2].
> > > > > >
> > > > > > 3: After all, the best way to justify your conclusion is to test your mixed-precision model on hardware platforms.
> > > > > >
> > > > > > [1]:  Analytical Guarantees on Numerical Precision of Deep Neural Networks:  http://proceedings.mlr.press/v70/sakr17a.html
> > > > > > [2]:  Uniq: Uniform noise injection for the quantization of neural networks.
> > > > > >  https://arxiv.org/pdf/1804.10969.pdf

---

> > > > > > > ### Author Response · Authors · 2018-12-06
> > > > > > > **Response**
> > > > > > >
> > > > > > > First of all, the "on-the-fly" memory consumption you mentioned is at most bi-linear (weight-bit x activation-bit), instead of exponential, with respect to precisions. A simple fact is, to multiply an M-bit weight with an N-bit activation, it involves M*N bit-wise multiplications (M*N bit memory footprint), and the result can be stored in a (M+N)-bit number. Unless your fixed point arithmetic is implemented by a LUT, exponential cost is by all means an over-estimation.
> > > > > > >
> > > > > > > Second, just to re-state my clarification in the previous response. We used two metrics in this work:
> > > > > > >
> > > > > > > 1/ We use a linear cost for MODEL SIZE (storage space) reduction. In fact, this is widely adopted in many previous works such as [1, 2]. Also, it is equivalent to the "representational cost" in [3], as the anonymous-2 pointed out.
> > > > > > >
> > > > > > > 2/ We use a bi-linear cost (weight-bit x activation-bit) for "COMPUTATIONAL COST" reduction. If the weight-bit is the same as the activation-bit, the metric is equivalent to the quadratic cost as in [3, 4]. It is also equivalent to your description of "on-the-fly" memory consumption.
> > > > > > >
> > > > > > > I agree that BOPs in [5] is a more precise bit-wise operation count by considering not only multiplications (weight-bit x activation), but also additions (weight-bit + activation-bit + constant), but that's dependent on the hardware implementation.
> > > > > > >
> > > > > > > As I stated in the previous response, we used the same metric to compute the model size and computational cost reduction rate for our method and previous baselines, so the results are directly comparable. You can also convert results from other methods to the same metric as ours, or our result to other metrics for comparison.
> > > > > > >
> > > > > > > Finally, I want to point out that a sensible choice of metric depends on the hardware implementation. The NAS framework proposed in our paper can easily adopt different metrics for different hardwares and search for different mixed precision strategies.
> > > > > > >
> > > > > > > [1] https://arxiv.org/abs/1602.07360
> > > > > > > [2] https://arxiv.org/abs/1510.00149
> > > > > > > [3]  Analytical Guarantees on Numerical Precision of Deep Neural Networks:  http://proceedings.mlr.press/v70/sakr17a.html
> > > > > > > [4] DoReFaNet: https://arxiv.org/abs/1606.06160
> > > > > > > [5] https://arxiv.org/pdf/1804.10969.pdf

---

### Meta-Review · Area_Chair1 · 2018-12-16
**Area chair recommendation**

**Confidence:** 4
**Recommendation:** Reject

**Metareview:**

The paper proposes a quantization framework that learns a different bit width per layer.  It is based on a differentiable objective where the Gumbel softmax approach is used with an annealing procedure.  The objective trades off accuracy and model size.

The reviewers generally thought the idea has merit.  Quoting from discussion comments (R4): "The paper cited by AnonReviewer 3 is indeed close to the current submission, but in my opinion the strongest contribution of this paper is the formulation from architecture search perspective."
The approach is general, and seems to be reasonably efficient (ResNet 18 took "less than 5 hours")

The main negatives are the comparison to other methods.  In the rebuttal, the authors suggested in multiple places that they would update the submission with additional experiments in response to reviewer comments.  As of the decision deadline, these experiments do not appear to have been added to the document.
In the discussion: R4: "This paper seems novel enough to me, but I agree that the prior work should at least be cited and compared to. This is a general weakness in the paper, the comparison to relevant prior works is not sufficient." R3: "Not only novel, but more general han the prior work mentioned, but the discussion / experiments do not seem to capture this."

With a range of scores around the borderline threshold for acceptance at ICLR, this is a difficult case.  On the balance, it appears that shortcomings in the experimental results are not resolved in time for ICLR 2019.  The missing results include ablation studies (promised to R4) and a comparison to DARTS (promised to R3): "We plan to perform the suggested experiments of comparing with exhaustive search and DARTS. The results will be hopefully updated before the revision deadline and the camera-ready if the paper is accepted." These results are not present and could not be evaluated during the review/discussion phase.